# Combined Self-Attention Mechanism for Chinese Named Entity Recognition in Military

**Fei Liao [1],\*, Liangli Ma [1], Jingjing Pei [2] and Linshan Tan [2]**

[1] College of Electronic Engineering, Naval University of Engineering, Wuhan 430033, China
[2] Force 91001, Beijing 100841, China
\* Correspondence: liaofei0209@163.com

**Abstract:** Military named entity recognition (MNER) is one of the key technologies in military information extraction. Traditional methods for the MNER task rely on cumbersome feature engineering and specialized domain knowledge. In order to solve this problem, we propose a method employing a bidirectional long short-term memory (BiLSTM) neural network with a self-attention mechanism to identify the military entities automatically. We obtain distributed vector representations of the military corpus by unsupervised learning and the BiLSTM model combined with the self-attention mechanism is adopted to capture contextual information fully carried by the character vector sequence. The experimental results show that the self-attention mechanism can improve effectively the performance of MNER task. The F-score of the military documents and network military texts identification was 90.15% and 89.34%, respectively, which was better than other models.

**Keywords:** military named entity recognition; self-attention mechanism; BiLSTM

## 1. Introduction

For the warring parties, how to extract a large amount of military information accurately in the shortest time will be the key link to win the initiative of war. Military named entity recognition (MNER) is a fundamental and important link of military information extraction used to detect military named entities (MNEs) from military text and classify them into predefined categories, such as troops, weapons, locations, missions, and organizations. It can extract valuable information from raw data to improve the efficiency of intelligence reconnaissance, command decision-making, organization, and implementation.

At present, military named entity recognition has a wide range of applications in the construction of a knowledge graph, intelligent question-answering systems, and in information retrieval. But the MNER task is still full of challenges due to the following reasons: (1) The limited number of supervised training data; (2) compared with English, Chinese lacks the explicit vocabulary boundary and the inherent definite article. At the same time, the proprietary vocabulary has no hint information such as spelling changes [1] (3). In addition to military instruments, other military texts do not have a uniform language expression format. The military corpus always has a large number of complex expressions, such as combination, nesting and abbreviation. Currently, most research on MNER has employed statistical methods of machine learning, including conditional random fields (CRF) [2], the hidden Markov model (HMM) [3], and maximum entropy (ME) [4], and so forth. These methods depend on handcrafted features and domain-specific knowledge resources excessively. However, the process of building feature templates is inefficient and expensive. And these traditional models have poor generalization ability.

To avoid cumbersome feature engineering and reduce the dependence on linguistic knowledge, our objective is to provide a neural network architecture that combines a bidirectional long short-term memory (BiLSTM) network with a self-attention mechanism to learn contextual features automatically. The experiment results show that our proposed model can improve the performance of MNER tasks.

The main contents of the paper are arranged as follows. Section 2 introduces the related work in the field of MNER briefly. Section 3 describes the model we proposed for MNER task in detail. Section 4 presents the experimental settings and process. Section 5 discusses the experimental results from multiple perspectives. Section 6 summarizes the research work of this paper and clarifies the direction of the next research.

## 2. Related Works

The concept of named entity recognition (NER) [5] was proposed at Message Understanding Conference-6, and the research work on NER has been developed for more than 20 years. Due to the inherent particularity and structural complexity of Chinese, Chinese NER research started later and was more difficult than English NER. In addition, MNEs have certain domain specificities and unique nomenclature, so the current research on identifying MNEs is not deep enough.

Traditionally, the MNER systems are based on either one of two methods: One is based on handcrafted rules and a dictionary; the other relies on statistical methods, such as CRF, HMM and support vector machine (SVM) [6]. The first method constructs the rules by linguistic experts manually and assigns weights to each rule, and then determines the type according to the conformity of entities and rules. Based on the rules and a dictionary, Guo et al. [7] adopted a custom semantic pattern to extract entities in tactical reports. Although the recognition accuracy was high, its compilation process was time-consuming. An approach based on rules is difficult to cover all the language phenomena and there are some problems such as poor portability, updating and maintenance difficulties. It requires linguistic experts to rewrite rules for different systems.

The statistical-based method mines and analyzes the linguistic information contained in the training corpus by statistical models, and extracts features from the training corpus, including word features, context features, and semantic features. Shan et al. [8] utilized the CRF model with a small granularity strategy to learn text features and then identified MNEs in operational documents. The authors in Reference [9] proposed a method based on rules and the CRF model to construct a semi-automatic MNER system. According to the grammatical characteristics of military text, the system established an efficient feature set to recognize MNEs. Furthermore, another statistical model conducted by Reference [10], when training the CRF model, external dictionary features were added to the model to make up for the shortage of training data and improve the accuracy of recognition to a certain extent. However, this type of statistical model has a common problem. Handcrafted features and domain-specific knowledge resources (e.g., a manually annotated dataset) are needed as inputs to train these models. The recognition effect is very dependent on the quality of the selected features.

With advances in deep learning, researchers have gradually applied the method of deep learning to high-performance MNER tasks, which can avoid cumbersome feature engineering. The deep learning method for sequence tagging is mainly to detect potential information in text by building a multi-layer neural network structure. Considering the constructive features of military equipment names, the author in Reference [11] put forward an identification method with the model of deep neural network based on the character of the word vector and state. In view of the difficulty of building feature templates in military texts, Zhu et al. [12] proposed an entity recognition framework based on BiLSTM-CRF. They combine the word and character vector as input to improve the performance of the model. The experimental results show that the proposed method is feasible and effective, and the F1 value on the test corpus reached 87.38%. Zhang et al. [13] adopted an architecture based on CNN-BiLSTM-CRF for identifying MNEs of combat documents. They used a convolutional neural network (CNN) to obtain the character embedding and achieve an F1-value of 89.89% on the test corpus.

## 3. BiLSTM-Self-Attention-CRF Model

We consider the MNER task as a sequential annotation problem like most NER tasks. In our work, we adopt a bi-directional LSTM with conditional random fields as a basic model for recognizing MNEs, which follow the best English NER model presented by References [14–16]. Unlike the mainstream framework of NER, we combine a self-attention mechanism into the basic model to acquire the relevant features of military texts in depth.

First, we convert the input sentences into a sequence of character vectors based on a pre-training language model, which was proposed by Mikolov et al. [17]. Then, character embedding is regarded as inputs to the BiLSTM encoder layer. BiLSTM learns to how to obtain contextual features for identifying MNES. The output of the encoder layer will be fed into self-attention layer before decoding. Finally, we employ the CRF model to yield the predicted label sequence.

Our neural network architecture is shown in Figure 1. In general, it is designed as an encoder-decoder architecture and can be roughly divided into four layers: (1) The embedding layer; (2) bi-LSTM encoding layer; (3) self-attention mechanism layer; (4) CRF decoding layer. In this section, we will introduce the proposed MNER model in detail combining with the characteristics of the military text.

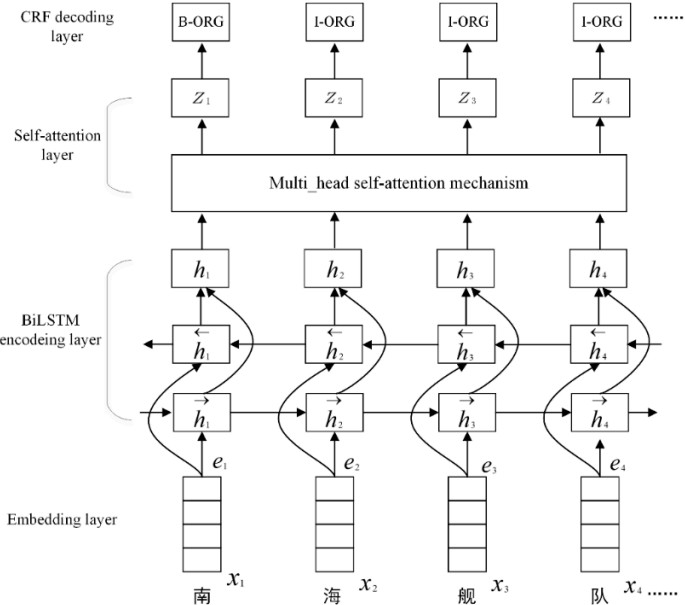

**Figure 1.** The architecture of BiLSTM-self_attention-CRF model. The input sentences, $(x_1, x_2, \ldots, x_n)$, are converted into a sequence of character vectors, $(e_1, e_2, \ldots, e_n)$. Character embedding is regarded as iuputs to BiLSTM encoder layer. BiLSTM learn to how to obtain contextual features for identifying MNES. The output of BiLSTM encoder layer will be fed into self-attention layer before decoding. Finally, we employ CRF model to yield the predicted label sequence.

### 3.1. Embedding Layer

The standard method of existing state-of-the-art models for English NER treats it as a word-by-word sequence labeling task [18]. English words can be quickly and clearly separated by explicit word separators, such as blank spaces. However, in the Chinese language, there is no natural separator between words, which results in word boundaries being ambiguous. The inaccuracy of Chinese word segmentation will affect the performance of subsequent entity recognition tasks. In order to avoid the potential problem of error accumulation caused by Chinese word segmentation, we adopt a character-based method to recognize Chinese MNEs, referring to the work of He et al. [19]. The authors in References [20,21] also prove that the character-based method performs better than the word-based method in the Chinese NER task.

Given a military document A including a sequence of sentences, $(s_1, s_2, \ldots, s_m)$, we set up $x = (x_1, x_2, \ldots, x_n)$ as the input sentence, where n represents the number of characters in the input sentence. We first need to map input sentences to vector sequence so that the neural network model can process the raw data simply. We use the distributed character-based representation proposed by Mikolov et al. [22], which maps sparse and high-dimensional one-hot codes to dense and low-dimensional character vectors through unsupervised training. In the first step, we use the skip-gram model of Word2vec to pre-train large-scale unlabeled Chinese corpus and get the character vector dictionary. Then, we map each Chinese character $x_i$ in the input sentence to a vector $e_i$ by querying the dictionary. By this way, input sentences are converted to character-based vector sequences and then these vector sequences will be entered to the BiLSTM encoding layer.

*3.2. Bi-LSTM Encoding Layer*

3.2.1. LSTM

Long short-term memory (LSTM) is a special class of recurrent neural network (RNN). It is designed to solve the problem of gradient disappearance or gradient explosion when processing long sequence data and better model the long-distance dependency relationship than general RNN [23]. The hidden layer of LSTM has three special gating mechanisms, including the forget gate, input gate, and output gate. Three gates can be used to control the update, attenuation, input, and output of information in memory cells. In short, the main idea of LSTM is to manage the information in the memory unit by learning the parameters of the three gates, so that useful information can be stored in the memory unit after a long sequence.

At a certain time $t$, the forget gate determines the amount of information discarded by the cell, which is given by:

$$f_t = \sigma(W_f \cdot h_{t-1} + U_f \cdot x_t + b_f) \tag{1}$$

where $x_t$ is the input of the current moment, $h_{t-1}$ is the output of the hidden layer at the last moment, $U_f$ represents the weight of the input information, $w_f$ represents the weight of the forget gate, and $b_f$ is the bias term. The input gate is to determine how much information the LSTM will save when the cell is updated, which is given by:

$$i_t = \sigma(W_i \cdot h_{t-1} + U_i \cdot x_t + b_i) \tag{2}$$

$$\widetilde{C_t} = \tan h(w_c \cdot h_{t-1} + U_c \cdot x_t + b_c) \tag{3}$$

$$c_t = f_t c_{t-1} + i_t \widetilde{C_t} \tag{4}$$

where $U_i$, $W_i$, $U_c$, and $W_c$ are the parameter matrices, $b_i$ and $b_c$ represent the bias parameters, $C_t$ is the current cell state, and $C_{t-1}$ is the state of the cell at the previous moment. The output gate ultimately determines the output vector of the LSTM model, which is given by:

$$o_t = \sigma(W_o \cdot h_{t-1} + U_o \cdot x_t + b_o) \tag{5}$$

$$h_t = o_t * \tanh(c_t) \tag{6}$$

where $W_o$ and $U_o$ are the parameter matrices, $b_o$ is the bias parameters, and $h_t$ is the final output of the current LSTM.

3.2.2. BiLSTM

LSTM only uses past information from text sequences to predict the current results. But for a NER task, both past and future information are beneficial for predictions. BiLSTM is composed of a forward LSTM and a backward LSTM. The BiLSTM model is better than a single LSTM in capturing context information in sentences, which can obtain the long dependence of sentences and the deep semantic expression of the text. So instead of a single forward network, we utilize BiLSTM to capture contextual

information from two directions. As shown in Figure 1, the network contains two sub-networks for the left and right sequence context, which are the forward and backward pass, respectively. A character embedding sequence of each word in a sentence, $(e_1, e_2, \ldots, e_n)$, is used as the input for each time step of bidirectional LSTM. Then, we use LSTM to compute the left context representations, $\overrightarrow{h_t} = (\overrightarrow{h_1}, \overrightarrow{h_2}, \ldots, \overrightarrow{h_n})$, and the right context representations, $\overleftarrow{h_t} = (\overleftarrow{h_1}, \overleftarrow{h_2}, \ldots, \overleftarrow{h_n})$. The final expression of each word is given by concatenating the left and right context representations, $H = (h_1, h_2, \ldots, h_n)$.

### 3.3. Self-Attention Layer

The greatest limitation of the traditional encoder-decoder structure is that the only connection between them is a fixed-length of the intermediate semantic vector. That is to say, the encoder compresses the whole sequence of information into a fixed-length vector. This approach has two drawbacks. One is that the semantic vector cannot fully represent the information of the entire sequence, which results in the information in the input sequence not being fully utilized. The other is, as the length of the input sequence grows, the information entered first may be overwritten by the information entered later. Recently, the self-attention mechanism has been widely used in image recognition, natural language understanding, and other fields to break the limitations of the traditional encoder-decoder structure, so as to extract as much feature information as possible from the input sequences [24]. To enhance the ability of the model to process information, we combine BiLSTM with a self-attention mechanism to capture lexical features and semantic information deeply. In the task of MNER, the semantic information of each Chinese character in a sentence does not have the same effect on this task. This mechanism can automatically focus on the specific Chinese characters that play a decisive role in entity recognition and capture the important semantic information in the input sequences, whilst reducing the focus on useless information.

The self-attention mechanism is a selective mechanism for allocating limited information processing capacity, which can be used to find the relationship between the internals of the sequence. It can selectively focus on some important information and give high weights to important information, while assigning smaller weights to other information received at the same time. The attention mechanism can be described essentially as a mapping relationship between a query and a series of key-value pairs. The output is computed as a weighted sum of the values, where the weight assigned to each value is computed by a compatibility function of the query with the corresponding key [25].

Considering the small scale of military text corpus and the existence of many irregular texts, it is necessary to extract more text semantic features from multi-angle and multi-level perspectives. The demand cannot be met by an attention mechanism alone. Therefore, this paper uses the multi-attention mechanism proposed by Vaswani et al. [25] in 2017 to improve the performance of MNER. Figure 2 is the structure of the multi-attention mechanism.

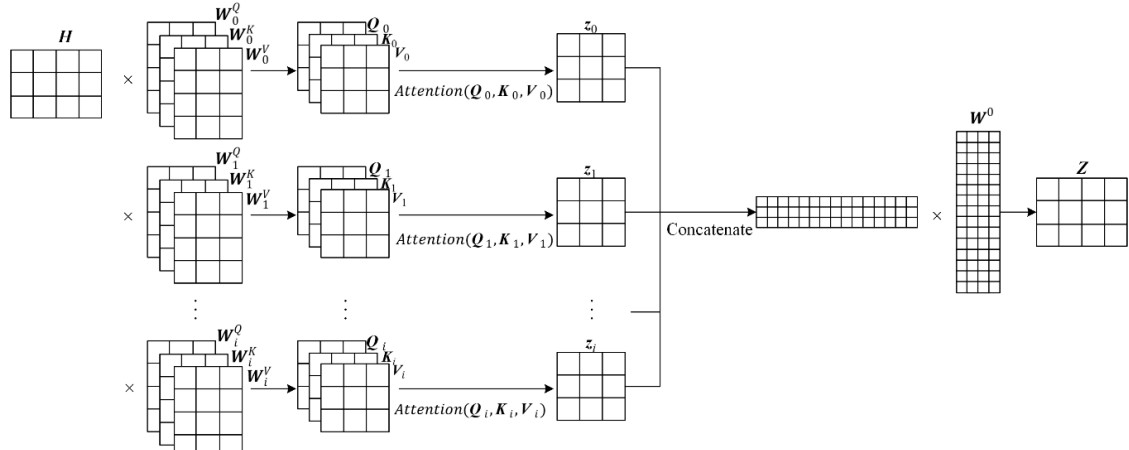

**Figure 2.** The structure of multi-attention mechanism.

For the complete hidden state sequence of the BiLSTM output, $H = (h_1, h_2, \dots, h_n)$, the vector matrix $H$ is mapped to the three weight matrices obtained by training, $w_i^Q$, $w_i^K$, $w_i^V$, for h times. The $Q$, $K$, and $V$ vectors obtained from each mapping are fed into the attention function to generate the output $z_i$. Finally, the output matrix $Z$ is obtained by splicing the h outputs together, so that more comprehensive feature information can be obtained.

The attention function here uses the scaled dot-product attention function, which is essentially an attention mechanism that uses point multiplication to calculate similarity [26]. The specific calculation formula is as follows:

$$attention(Q, K, V) = softmax\left(\frac{QK^T}{\sqrt{d_k}}\right)V \tag{7}$$

The first step is to compute the dot products of the query with all keys to get the similarity scores between the character vectors. The second step is to divide by $\sqrt{d_k}$, which leads to having more stable gradients. The third step is applying a softmax function to calculate the weight coefficient. Finally, according to the weighted coefficients, a weighted summation operation is performed on the value to get an attention output. The multi-attention mechanism is composed of several scaled dot-product attentions. Each attention focuses on different parts of the input information, learning semantic information from different dimensions and representation spaces. The specific calculation formula is as shown in Equations (8) and (9).

$$head_i = attention\left(HW_i^Q, HW_i^K, HW_i^V\right) \tag{8}$$

$$multi\_head\ (Q,K,V) = concat\ (head_1, head_2, \dots, head_n) \tag{9}$$

### 3.4. CRF Decoding Layer

In general, there are two methods to estimate the current labels. The first method is to make tagging decisions independently by using the softmax classifier. The softmax classifier is a normalized exponential function, where each position to be labeled is treated as a sample and classified independently. However, a single-point classification will result in the great loss of information and wrong annotation, because there is a strong dependence between labels. When we imposed a label for each Chinese character, the format of the labeling strategy for the NER tasks has strong constraints, for instance, the "B-Weapon" tag cannot follow behind the tag "I-Weapon". Hence, it is insufficient that the softmax classifier acts as an output layer.

CRF is a probability graph model that follows the Markov property, which focuses on the sentence level rather than decoding the tag separately. We use a chain conditional random field to decode the information vectors generated by the BiLSTM and self_attention layers. This model is widely used in the sequential annotation problem and it can model the whole implicit state.

Given an input sequence $x = (x_1, x_2, \dots, x_n)$, the score of prediction sequence, $y = (y_1, y_2, \dots, y_n)$, is computed as:

$$score(x, y) = \sum_{i=0}^{n} Z_{i,y_i} + \sum_{i=0}^{n} A_{y_i, y_{i+1}} \tag{10}$$

where $Z$ is the output matrix of the self-attention layer, and $Z_{i,y_i}$ corresponds to score of label $j$ for character $i$. $A$ represents the matrix of transition scores and $A_{y_i, y_{i+1}}$ corresponds to the transition score of label $y_i$ to $y_{i+1}$. Then, the normalized probability can be obtained by using Softmax. We denoted $Y_x$ as the set of the possible label sequences.

$$P(y|x) = \frac{\exp(score(x, y))}{\sum_{y' \in Y_x} \exp(score(x, y'))} \tag{11}$$

In the process of model training, the logarithmic likelihood probabilities of the correct label-sequence are maximized. We use Viterbi algorithm [27] to get the highest score tagging

sequence in all sequences in the final decoding stage, and take the global optimal sequence as the final result of military named entity recognition, and the model training is completed.

## 4. Experiments

### 4.1. Data Sets

Due to the lack of authoritative and unified military corpus, we employed a manually annotated military NER corpus to train and evaluate our models, which was created by Naval Big Data Research and Application Center from multiple sources. The military corpus can be divided into two parts: The military documents data set (MDD) and the network military texts data set (NMTD). MDD was selected from the exercises and duty compilation materials of a certain force for nearly three years. There are 201 military documents in MDD, including 96 combat documents, 105 duty documents. NMTD was crawled and compiled from large military websites such as "China Military Network" and "Global Military Net" through web crawler tools. There are 164 network military texts in NMTD, including 112 internet military news and 52 military blogs. Four types of MNEs were labeled in the experiment, including weapon (WEA), location (LOC), organization (ORG) and mission (MIS). In addition, the MNE labels were annotated using the BIO annotation strategy. B represents the beginning of the entity, while I represents the character inside of the entity, and O represents the other non-entity characters. In order to validate the performance of our model, we took 80% from the dataset as the training set and the remaining 20% as the test set. The details of the two datasets are shown in Tables 1 and 2.

**Table 1.** The details of military documents data set (MDD).

| Type | Sentences | Characters | MNEs |
|---|---|---|---|
| data set | 4857 | 175426 | 826 |
| train | 3885 | 139652 | 654 |
| test | 972 | 35774 | 172 |

**Table 2.** The details of network military texts data set (NMTD).

| Type | Sentences | Characters | MNEs |
|---|---|---|---|
| data set | 6849 | 207538 | 1135 |
| train | 5478 | 165937 | 891 |
| test | 1371 | 40601 | 244 |

### 4.2. Environment and Settings

The software and hardware environment for our experiment is shown in Table 3. In the experiment, we chose the skip-gram model in word2vec to train the character vector. Table 4 is the hyperparameter settings of our experiment.

**Table 3.** The environment settings.

| Type | Settings |
|---|---|
| OS | Ubuntu 18.04 |
| CPU | i7-8700K 3.70GHz |
| GPU | NVIDIA TITAN Xp (12GB) |
| Tensorflow | 1.12.0 |
| python | 3.6 |

**Table 4.** The hyperparameter settings of our experiment.

| Hyperparameter name | value |
|---|---|
| Character embedding size | 100 |
| Window size | 5 |
| BiLSTM hidden size | 200 |
| Dropout | 0.5 |
| Maximum number of epochs | 64 |
| Initial learning | $10^{-4}$ |

*4.3. Experimental Baselines*

For the MNER task, many researchers have proposed many different methods to solve it. We selected some of the research work accomplished previously and compared them, on the premise of using the same datasets and evaluation indicators. We set up multiple sets of contrast experiments for different types of military text to verify the effectiveness of our proposed method. The following models are regarded as the baseline of our experiment:

- The work using CRF carried out by Yuntian Feng in Reference [10];
- The approach relying on BiLSTM-CRF model proposed by Reference [12];
- The author in Reference [13] put forward a method based on CNN-BiLSTM-CRF for MNER.

*4.4. Evaluation Metrics*

The performance of MNER was estimated by three types of indicators in this experiment, namely, precision (P), recall (R), and the F-score (F1). Simultaneously, the F1 value is a weighted average of P and R, which provided a comprehensive evaluation of the performance of the model. The calculation formulas of the three evaluation indicators are as follows:

$$P = \frac{\text{number of correctly recognized entities}}{\text{number of identified entities}} \times 100\% \tag{12}$$

$$R = \frac{\text{number of correctly recognized entities}}{\text{number of entities in the sample}} \times 100\% \tag{13}$$

$$F_1 = \frac{2 \times P \times R}{P + R} \tag{14}$$

**5. Results and Discussions**

We comply our experiment on the MDD and NMTD described above, respectively. Table 5 lists the experimental results obtained by different models. From the results in Table 5, the performance of our proposed model with self-attention mechanism was better than previous baseline models. The F1 values of our model for the identification of military documents and network military texts were 90.15% and 89.34%, respectively.

**Table 5.** The experimental results obtained by different models.

| Number | Model | Data Set | Samples | Identify | Correct | P | R | F1 |
|--------|-------|----------|---------|----------|---------|---|---|-----|
| 1 | CRF [10] | MDD | 826 | 819 | 600 | 73.26% | 72.60% | 72.93% |
| | | NMTD | 1135 | 1075 | 772 | 71.82% | 68.02% | 69.87% |
| 2 | CNN-BiLSTM-CRF [13] | MDD | 826 | 824 | 734 | 89.05% | 88.87% | 88.96% |
| | | NMTD | 1135 | 1130 | 993 | 87.94% | 87.52% | 87.73% |
| 3 | BiLSTM-CRF [12] | MDD | 826 | 802 | 700 | 87.30% | 84.78% | 86.02% |
| | | NMTD | 1135 | 1110 | 952 | 85.78% | 83.86% | 84.81% |
| 4 | **BiLSTM-Self_Att-CRF** | MDD | 826 | 798 | 732 | 91.72% | 88.63% | **90.15%** |
| | | NMTD | 1135 | 1115 | 1005 | 90.13% | 88.56% | **89.34%** |

Note: "samples" represents the total number of military entities in the samples. "Identify" is the total number of identified military entities. "Correct" denotes the total number of correctly identified military entities.

## 5.1. Comparison of Different Architecture

Based on the mainstream architecture of BiLSTM-CRF, we add a multi-attention mechanism to the military named entity recognition model. In order to verify that the model combining with the multi-attention mechanism could improve the recognition effect, we trained BiLSTM-CRF (model 3) and BiLSTM-Attention-CRF models (our proposed model) on the training corpus, respectively, and then recognized the MNEs of the military documents and network military texts in the test corpus separately. As can be seen from Table 5, the F1 scores of the BiLSTM-CRF model for the identification of military documents and network military texts were 86.02% and 84.81%, respectively. And our proposed model achieved 90.15% and 89.34% in the F1 value, which was higher than the general BiLSTM-CRF model. The experimental results showed that using the self-attention mechanism to solve MNER tasks could maximize the use of effective features in resources and reduce the interference of useless features, and better capture context-related information and fully mine the features of the text itself, thus improving the performance of MNER.

## 5.2. Comparison with Other Models

For the purpose of further evaluating our system, the performance of our proposed model is compared with those of other existing state of the art methods. We have reproduced the models in References [10,12,13] using the same datasets and evaluation indicators.

Overall, the performance of model 1 was significantly worse than models 2, 3, and 4. The reason is that model 1 uses a traditional method based on CRF, which heavily relies on hand-crafted features and domain language knowledge. It has certain limitations and drawbacks, because custom feature templates do not cover all semantic representations, resulting in missing information, which is useful for recognition. On the contrary, models 2, 3, and 4 are based on a neural network. Both of them adopt RNN for sequence labeling to learn language features automatically without manual intervention. The results emphasize that employing BiLSTM leads to achieve excellent performance and improve MNER tasks.

Comparing model 2 and our model, in the same case using the BiLSTM neural network, model 2 uses a convolutional neural network to improve the input of BiLSTM and our model extended the output of BiLSTM by the self-attention mechanism. The F1 values of model 2 were 88.96% and 87.73%, while the results of our model increased by 1.19% and 1.61%, respectively. In the MNER task, the self-attention mechanism was better at processing information than the convolutional neural networks.

## 5.3. Different Types of Datasets

By analyzing the above multiple sets of comparative experiments, we found that the performance of military document recognition was always slightly higher than the performance of network military text recognition. The main reason is that military documents have a uniform format and writing

norms, and the boundary of the MNEs is obvious. On the contrary, network military texts are often mixed with a lot of interference information, and there are more colloquial and networked irregular vocabularies, which poses great difficulties for identification tasks.

## 6. Conclusions

In this paper, we utilized a character-based neural network, based on BiLSTM-self-attention-CRF for a Chinese military named entity recognition task, which did not rely on cumbersome features engineering and any knowledge sources. We obtained distributed vector representations of the military corpus by unsupervised learning, and the self-attention mechanism and BiLSTM model were combined to capture contextual information. In the experiments conducted, we trained different models for identifying the military entities in the military documents and network military texts. The results showed that the performance of our proposed model was better than baseline models. The F1 values of our model for the identification of military documents and network military texts were 90.15% and 89.34%, respectively. Further work considering the use of external resources and multi-task joint learning to improve the performance of military named entity recognition task is required.

**Author Contributions:** Conceptualization, F.L. and L.M.; methodology, F.L.; validation, F.L., and L.M.; formal analysis, F.L.; investigation, F.L.; resources, J.P.; data curation, F.L. and L.T.; writing—original draft preparation, F.L.; writing—review and editing, F.L. and L.T.; supervision, J.P.; project administration, L.T.

**Funding:** This research was funded by the National Natural Science Foundation of China (grant number 61802425).

**Conflicts of Interest:** The authors declare no conflict of interest.

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
