# Peer review of "Combined Self-Attention Mechanism for Chinese Named Entity Recognition in Military"

_futureinternet, doi:10.3390/fi11080180_

Round 1

Reviewer 1 Report

This paper presents interesting technique which is utilized for NER in Military specifically for Chinese language.
However, I put some suggestion to escalate this paper's quality.

Some statements are confusing in term of its English. All authors must revisit all statements.

Related works are well presented in brief and most of them are new. However, all abbreviations need to be described before they are continually used.

The important point to be highlighted is: why BiLSTM and self attention mechanisms are chosen? There is no justification regarding that choices. The authors must put it inside the manuscript.   

Where did the language model in line 99 come from? Is it self produced or not? If it is self-produced, the author needs to describe it.

It comes to confuse the reader when all things in figure 1 are not explained in the text.

Line 199 mentions that the authors used multi-attention mechanism while the paper title and figure 2's title are self attention mechanism. Which one si correct?

The justification of result is clear and very well presented. However, I suggest to elaborate the indicators of how did the attention mechanism work to support in boosting up the performance.

External resources which are mentioned in line 344 are not clear. What is the author's objective to mention this part? The authors may elaborate more regarding this statement.

Reviewer 2 Report

The authors propose an approach for Named Entity Recognition in the military domain, that uses BiLSTM and self-attention. The topic in interesting and worth investigating and the paper is well written.

The authors should better highlight how the proposed approach is different from existing ones, such as the one described in https://www.aclweb.org/anthology/D18-1017, which also uses BiLSTM+CRF+self-attention. 

# Other recommendations

At line 98, the paper mentions that "vectors based on pre-training language model".  The authors should state how and using what data have the vectors been pre-trained.

Round 2

Reviewer 2 Report

The authors have addressed the comments in the previous review.